# Anti-Sclerostin Antibodies in Osteoporosis and Other Bone Diseases

**DOI:** 10.3390/jcm9113439

**Published:** 2020-10-26

**Authors:** Stéphanie Fabre, Thomas Funck-Brentano, Martine Cohen-Solal

**Affiliations:** Department of Rheumatology, Lariboisière Hospital, INSERM U1132 and University of Paris, 75010 Paris, France; stephanie.fabre@inserm.fr (S.F.); thomas.funck-brentano@aphp.fr (T.F.-B.)

**Keywords:** bone, fracture, osteoporosis, Wnt, sclerostin

## Abstract

The Wnt pathway is a key element of bone remodeling; its activation stimulates bone formation and inhibits bone resorption. The discovery of sclerostin, a natural antagonist of the Wnt pathway, promoted the development of romosozumab, a human monoclonal antibody directed against sclerostin, as well as other anti-sclerostin antibodies. Phase 3 studies have shown the efficacy of romosozumab in the prevention of fractures in postmenopausal women, against placebo but also against alendronate or teriparatide and this treatment also allows bone mineral density (BMD) increase in men. Romosozumab induces the uncoupling of bone remodeling, leading to both an increase in bone formation and a decrease in bone resorption during the first months of treatment. The effect is attenuated over time and reversible when stopped but transition with anti-resorbing agents allows the maintenance or reinforcement of BMD improvements. Some concerns were raised about cardiovascular events. Therefore, romosozumab was recently approved in several countries for the treatment of severe osteoporosis in postmenopausal women with high fracture risk and without a history of heart attack, myocardial infarction or stroke. This review aims to outline the role of sclerostin, the efficacy and safety of anti-sclerostin therapies and in particular romosozumab and their place in therapeutic strategies against osteoporosis or other bone diseases.

## 1. Introduction

Bone is a complex and active tissue in constant remodeling. Each remodeling cycle is initiated by a phase of resorption of the mineralized matrix by osteoclasts, followed by a phase of bone formation by osteoblasts [1]. These two processes are temporally and spatially coupled in order to maintain skeletal integrity [2]. A better understanding of the regulatory pathways involved in bone remodeling has enabled the identification of new therapeutic targets for the treatment of low bone mass and prevention of fractures. Among them, the canonical Wnt (wingless-related integration site) pathway plays a central role, stimulating bone formation and inhibiting bone resorption [3]. Sclerostin is one of the natural Wnt pathway’s antagonists [4] and its loss of function is responsible for rare bone diseases characterized by a high bone mass [5,6,7,8], which has motivated extensive research on this molecule. Several monoclonal antibodies directed against the human form of sclerostin and inhibiting its action have been developed [9,10,11]. In particular, romosozumab has been evaluated in several phase 3 studies in postmenopausal and male osteoporosis, but other anti-sclerostin antibodies are currently assessed in various skeletal disorders. This review aims to outline the role of sclerostin, the efficacy and safety of anti-sclerostin therapies and in particular romosozumab and their place in therapeutic strategies against osteoporosis or other bone diseases.

## 2. Sclerostin, a Molecule That Regulates Bone Remodeling

Sclerostin is a natural inhibitor of the canonical Wnt pathway (Figure 1a). The signaling pathway is one of the key elements of bone formation; it is involved in the commitment of mesenchymal stem cells to the osteoblastic lineage and their differentiation into osteoblasts at different stages. Besides, the pathway activation also induces an inhibition of bone resorption by osteoclasts, through the induction of the expression of osteoprotegerin (OPG, a decoy receptor for RANK-Ligand) by osteoblasts and osteocytes [12,13]. Wnt molecules also have a direct action on osteoclasts, stimulating the proliferation of osteoclast progenitors but inhibiting their differentiation [14,15,16,17]. In the absence of Wnt ligand, a destruction complex phosphorylates beta-catenin, targeting it for ubiquitination and degradation by the proteasome. When Wnt ligands bind to Frizzled receptor and LRP5/6 (low density lipoprotein receptor-related protein 5 or 6) co-receptor, there is an inhibition of the destruction complex. Beta-catenin accumulates, translocates into the cell nucleus and triggers the transcription of genes involved in bone formation (Figure 1b). Non-canonical Wnt pathways stimulating bone formation can also be activated by Wnt ligands but they increase bone resorption [3] (Figure 1c). The Wnt pathway is physiologically regulated by natural antagonists, such as DKK1 (Dickkopf 1) or sclerostin, encoded by the *SOST* gene [18]. By binding to the LRP5/6 co-receptor, sclerostin inhibits the canonical Wnt pathway, and therefore bone formation (Figure 1a).

Sclerostin is mainly produced by cells within mineralized matrices and in particular mature osteocytes, but also hypertrophic chondrocytes or cementocytes [19,20]. It is also expressed in other tissues, mainly the kidney, liver, heart, or carotid arteries and by non-mineralized cells in pathological conditions [21]. Several factors regulate the expression of sclerostin by osteocytes, such as mechanical stress, parathyroid hormone or cytokines of the IL-6 family [22]. Sclerostin expression increases with aging [23]. Identification of the Wnt signaling pathway and discovery that a loss of function of sclerostin is responsible for high bone mass diseases, such as sclerosteosis or Van Buchem disease, have promoted sclerostin as a potential therapeutic target for increasing bone formation, in osteoporosis or other bone diseases.

## 3. Efficacy of Romosozumab in Osteoporotic Women and Men

Research has led to the development of romosozumab, a monoclonal antibody that binds to the human form of sclerostin and inhibits its action (Figure 1b) [24], thereby promoting the activation of the Wnt pathway. Several phase 3 studies have shown the effectiveness of this treatment in postmenopausal osteoporosis.

The Fracture Study in Postmenopausal Women with Osteoporosis (FRAME) evaluated the efficacy of subcutaneous romosozumab administered 210 mg monthly for 12 months versus placebo, in 6390 patients suffering from postmenopausal osteoporosis [9]. The gain in bone mineral density (BMD) at the lumbar spine was 13.3% higher than in the placebo group; it reached at least 3% in 96% of romosozumab-treated patients and at least 10% in 68% of treated women while the percentage of patients who gained BMD in the placebo group was lower (22% and 1% respectively) [25]. At the femoral neck and total hip, the average gain in BMD was 5.9% and 6.9% greater than that on placebo, respectively. The BMD gain was above 3% in 78% of patients and above 10% in 16% of patients compared to 16% and 0% respectively in the placebo group). However, the effect of romosozumab on BMD is reversible when the treatment is discontinued. McClung et al. showed in a phase 2 study evaluating the efficacy of romosozumab over 24 months that in the absence of transition to another anti-osteoporosis treatment, patients returned to BMD values close to those measured at baseline approximately 1 year after cessation of treatment [26].

The effect on fracture incidence is consistent with the effect on BMD. After 12 months of romosozumab administration, vertebral fractures were reduced by 73% (incidence 0.5% vs. 1.8% for placebo). This effect was rapid since only 2 of the 16 vertebral fractures noticed in the romosozumab group were observed after 6 months of treatment. Regarding clinical vertebral fractures, there was a new or worsening fracture in 20 patients, 3 in the romosozumab group (incidence <0.1%) and 17 in the placebo group (incidence 0.5%). The 3 clinical vertebral fractures in the romosozumab group occurred within 2 months of treatment initiation [27]. The incidence of non-vertebral fractures was lower in the romosozumab group (1.6% vs. 2.1%), but this reduction was not significant (hazard ratio 0.75 [95% CI 0.53–1.05]). This could be due to a low risk of peripheral fracture in Latin American populations, which represent 43% of the FRAME study population (incidence of 1.2% in the placebo group for Latin America and 2.7% for the rest of the world). A post-hoc analysis showed that patients recruited in South America had less frequently a history of non-vertebral fracture [28] and that their probability of major osteoporotic fracture or hip fracture at 10 years, calculated using the Fracture risk assessment tool (FRAX) at baseline, was halved compared to that calculated for patients from the rest of the world. When patients from Latin America where excluded from the analysis, the risk of peripheral fracture was reduced by 42% in the romosozumab group, compared to the placebo group.

The anti-fracture efficacy of romosozumab may likely be due to the gain in BMD, but also to the improvement of bone structure. In monkeys, romosozumab improved bone architecture and bone strength, while maintaining bone quality and mineralization. Cortical bone was preserved without any increase in cortical porosity [29]. These results were confirmed in a subgroup of 107 patients of the FRAME study in which an improvement in bone microarchitecture (trabecular thickness and connectivity) was observed after 12 months, with no increase in cortical porosity [30].

Romosozumab was also evaluated versus placebo in men in the phase 3 BRIDGE study. It included 245 men aged 55 to 90 years old, with a T-score at the lumbar spine, total femur or femoral neck ≤ −2.5 SD or ≤−1.5 SD with a history of fragility fracture [31]. The gain in BMD at the lumbar spine and total femur was significantly higher in the romosozumab group than in the placebo group at 12 months (12.1% vs. 1.2% and 2.5% vs. −0.5%, respectively, *p* < 0.001). No fracture data are available, given the small number of patients, but in a pre-clinical model of male osteoporosis, anti-sclerostin therapy resulted in an improvement of microarchitecture and bone strength [32].

Real world data in Japan in a limited number of patients suggest that the effect of romosozumab, in terms of BMD gain, can be attenuated by prior use of another anti-osteoporosis treatment (teriparatide, denosumab or bisphosphonate), at least at the lumbar spine and total hip [33].

## 4. Mechanism of Action of Anti-Sclerostin Antibodies

Romosozumab induces a rapid increase in the marker of bone formation procollagen type I N-terminal propeptide (P1NP), which is followed by a slow decrease towards baseline levels, after 9 months of treatment in the FRAME study. Conversely, the markers for bone resorption β-CTX (C-terminal telopeptides of type I collagen) decrease rapidly and then reach a plateau around −50% of the initial level [9]. In a subgroup of patients, histological analysis of 34 transiliac bone biopsies showed that bone formation indices were indeed doubled after 2 months of treatment with romosozumab compared to placebo and that the resorption parameters were significantly reduced at the trabecular and endocortical levels [30]. Evolution of the histological and biochemical markers illustrates the uncoupling of bone formation and resorption observed at the onset of treatment with romosozumab, which is unique and distinguishes it from current anti-osteoporosis treatments (Table 1). This is possible through an increase in modeling-based bone formation [34], usually mainly seen during growth and in response to mechanical loading [35]. Unlike bone remodeling, modeling does not require a prior bone resorption step, but can be initiated de novo on a quiescent bone surface. Indeed, histomorphometric analysis using an osteoblast tracking technique in murine models and transcriptional profiling of osteoblastic subpopulations suggested that anti-sclerostin antibodies can increase bone formation by activating the bone lining cells, promoting their transition from quiescent to active matrix producing-osteoblasts [36,37]. In parallel, there is a decrease in bone resorption as remodeling surfaces are reduced and an increase in osteoblastic efficiency at remodeling sites [38]. The reduction of bone resorption can be explained by a lower RANKL/OPG ratio. However, a decrease of *Csf1* expression (colony stimulating factor 1), essential for osteoclast differentiation and survival, in osteoblasts and osteocytes, as well as an increase in *Wisp1* (Wnt1 inducible signaling pathway protein 1) expression, a negative regulator of bone resorption, are also probably implicated [29,39]. We currently do not have any data regarding a potential direct effect of sclerostin antibody therapy on the osteoclast lineage.

However, after approximately 6 to 12 months of treatment, P1NP levels have returned to baseline values and β-CTX still plateau at around −50% of the initial level, with ongoing bone mass accrual, though attenuated compared to the first months of treatment. Histological analysis of 73 transiliac bone biopsies after 12 months of romosozumab showed that both trabecular bone formation and resorption were reduced, indicating an overall decrease in bone turnover, as can be seen with anti-resorbing agents [30]. Self-regulation of bone formation after prolonged administration can be explained by a progressive decrease in the number of osteoblastic progenitors [38], possibly resulting from a decrease in their proliferation [42], as well as a decline in the number of osteoblasts and bone formation rate [39]. Activation of signaling pathways that limit mitogenesis and the progression of the cell cycle is probably involved [39]. Of note, this sequential effect appears later in cortical bone than in trabecular bone, likely because of the extent of trabecular and cortical bone surfaces or because of different regulation mechanisms in the two bone compartments [43]. However, the remodeling bone balance remains positive, which can be explained by ongoing increased osteoblastic efficiency at remodeling sites, reduced remodeling surfaces and resorption depth [38].

In mice, repeated administration of anti-sclerostin antibodies was associated with increased expression of inhibitors of the Wnt pathway, including sclerostin and DKK1, which may contribute to the attenuation of the response to treatment [44]. Thus, bi-therapy with concomitant anti-sclerostin and anti-DKK1 drugs have been suggested [22], or the use of a bispecific antibody against these two targets, which has shown interest in rodents and monkeys [45]. This warrants further studies, in particular in terms of toxicity. Attenuation of anti-sclerostin therapy effect over time raises the question of the modality of such treatment in the future.

## 5. Comparison with Other Anti-Fracture Treatments

The effect of romosozumab has also been evaluated either in comparison or in combination with current anti-fracture treatments.

### 5.1. Comparison with Teriparatide

Romosozumab was compared with teriparatide in patients naive to anti-osteoporosis treatment in a phase 2 trial [46]. The mean increase in BMD at 12 months was significantly higher in the group treated with romosozumab at a dose of 210 mg monthly, at the lumbar spine (11.3% vs. 7.1% in the teriparatide group, *p* ≤ 0.001), at the total hip (4.1% vs. 1.3%, *p* ≤ 0.001) or at the femoral neck (3.7%. vs. 1.1%, *p* ≤ 0.001). Subsequently, the phase 3 STRUCTURE trial compared the 2 treatments in 436 patients aged 55 to 90 with postmenopausal osteoporosis with a history of fracture and who received a bisphosphonate for at least 3 years before inclusion, and alendronate during the year preceding the inclusion [47]. The mean gain in BMD was greater in the romosozumab group at the lumbar spine, total hip and femoral neck at 12 months (9.8%, 2.9%, and 3.2% respectively vs. 5.4%, −0.5%, and −0.2% in the teriparatide group). Analysis of the hip by quantitative computed tomography (QCT) revealed that the greater increase in BMD observed on romosozumab was mainly related to the cortical compartment, the evolution of the trabecular volumetric BMD being similar in the two groups. Romosozumab also improved hip strength, estimated by finite element analysis (change from baseline of 2.5% vs. −0.7% for teriparatide, *p* < 0.0001). Regarding bone remodeling markers, P1NP levels were higher at 1 month, equal at 2 months and then lower in the romosozumab group compared to the teriparatide group. In parallel, CTX levels decreased at the onset of treatment and then returned to baseline values at 3 months, while they were increased from baseline values throughout the study in the teriparatide group. These results highlight the different modes of action of the 2 molecules. Indeed, teriparatide induces an increase in bone formation and resorption with a final positive bone balance, while romosozumab promotes initial bone modeling with a decrease of bone resorption, followed by a decrease in bone remodeling as described above. The effect of concomitant therapy with romosozumab and teriparatide has also been evaluated in rats [48]. This dual treatment was more effective than each treatment alone in improving BMD, bone microarchitecture, and bone strength at the distal femur and in increasing P1NP and osteocalcin, over a period of 12 weeks.

### 5.2. Comparison with Bisphosphonates

The ARCH study compared romosozumab to weekly alendronate for 12 months in 4093 patients with postmenopausal osteoporosis with a history of fragility fracture [49]. The gain in BMD was greater in the romosozumab group at the lumbar spine (13.7% vs. 5.0% for alendronate), as well as at the total hip (6.2% vs. 2.8%) and at the femoral neck (4.9% vs. 1.7%). The risk of new vertebral fracture was reduced by almost 40% with romosozumab compared to alendronate (4.0% vs. 6.3% respectively, *p* = 0.003), clinical fractures were reduced by 28% (3.9% vs. 5.4%, *p* = 0.027) and the risk of major non-vertebral osteoporotic fracture was reduced by 33% (2.9%, vs. 4.3%, *p* = 0.019).There was a trend towards a decrease in the non-vertebral (26%) and hip (36%) fractures, but this was not significant. A subanalysis conducted among the ARCH East-Asian population on 275 patients showed similar outcomes [50].

### 5.3. Comparison with Denosumab

To date there is no study that directly compares romosozumab with denosumab, but Cosman et al. have indirectly compared these two treatments based on their pivotal studies, FRAME and FREEDOM, whose populations have similar characteristics [25]. After one year of romosozumab, the gain in T-score was similar to that observed after 4.5 years of denosumab at the lumbar spine and after 3 years at the hip. After the two-year treatment sequence (12 months of romosozumab and then 12 months of denosumab), the gain in T-score at both sites was comparable to the one obtained after 7 years of denosumab. The results remained similar after adjustment for baseline characteristics of the two populations such as initial BMD, age, history of vertebral fracture. The fracture rate observed during the second year of the FRAME study (treatment with denosumab) was lower in the group of patients who had previously received romosozumab compared to those who had received placebo treatment, with a relative risk of fracture reduced by 81% for the spine (*p* < 0.001), and a trend towards a decrease for the other bone sites, although not significant.

## 6. Side Effects of Anti-Sclerostin Therapy

### 6.1. General Adverse Events

In the FRAME study, 7 patients presented a serious adverse reaction potentially related to hypersensitivity in the romosozumab group [9]. Reactions at the injection site, mostly of mild severity, were reported in 5.2% of patients in the romosozumab group and 2.9% in the placebo group. Osteonecrosis of the jaw was observed in two patients in the romosozumab group, the first after 12 months of treatment in a context of ill-fitting dentures, the second after 12 months of romosozumab and an injection of denosumab following a tooth extraction and subsequent osteomyelitis of the jaw. One atypical femoral fracture has been reported 3.5 months after the first dose of romosozumab, but pain had started before inclusion. In addition, binding anti-romosozumab antibodies developed in 18% of romosozumab patients during the first 15 months, and neutralizing antibodies were detected in 0.7% of patients, without effect on the efficacy or safety of the treatment. 

The frequency of adverse events observed in the STRUCTURE study [47] was generally balanced in the romosozumab and teriparatide groups. The main events were nasopharyngitis (13% vs. 10% respectively), hypercalcemia (<1% vs. 10%), arthralgia (10% vs. 6%), injection-site reactions (8% vs. 3%). Serious adverse events have been reported in 8% of patients on romosozumab and 11% of patients on teriparatide, but none were found to be related to treatment. Adverse events lead to discontinuation of treatment in 3% of patients on romosozumab and 6% on teriparatide. Atrial fibrillation occurred in two patients on romosozumab (<1%). In the ARCH [49] trial, which compared romosozumab to alendronate, side effects were overall similar in the two groups. However, in the first year more serious cardiovascular adverse events were observed in the romosozumab group (2.5% vs. 1.9%). These included in particular cardiac ischemic events (16 patients (0.8%) in the romosozumab group vs. 6 (0.3%) in the alendronate group; OR 2.35 [95% CI 1.03–6.77]), cerebrovascular events (16 (0.8%) vs. 7 (0.3%); OR 2.27 [95% CI 0.93–5.22]) and deaths (17 (0.8%) vs. 12 (0.6%). Heart failure and peripheral vascular ischemic events not requiring revascularization were lower in the romosozumab group. Cardiovascular risk factors were equally distributed between the two groups at baseline. In the BRIDGE study [31], romosozumab was generally well tolerated, but serious cardiovascular events were also more frequent in the romosozumab group than in the placebo group (4.9% vs. 2.5%). These included ischemic events (1.8% vs. 0%), cerebrovascular events (1.8% vs. 1.2%), and heart failure (0.6% vs. 0%), as well as one cardiovascular death in each group.

### 6.2. Cardiovascular Risk

The increased risk of serious cardiovascular events observed in the ARCH study comparing romosozumab to alendronate, which was not observed in the FRAME pivotal study or in the phase 2 trial comparing romosozumab to alendronate [46], raises questions. One possible explanation could be a protective role of alendronate. Indeed, some studies suggest that bisphoshonates may have a beneficial effect on the lipid profile, reducing the process of atherosclerosis and vascular calcifications but these results remain controversial [51,52,53]. Two recent meta-analyzes have failed to confirm the protective role of bisphosphonates regarding cardiovascular events [53,54]. Another hypothesis is that sclerostin plays a beneficial role in the cardiovascular system. Indeed, the Wnt pathway is also involved in cardiovascular remodeling and the inhibition of sclerostin could promote vascular calcification and the risk of cardiovascular event. However, a greater risk of vascular calcification has not been reported in sclerostin KO mice or in patients suffering from sclerosteosis or Van Buchem disease, related to a decrease in sclerostin expression [55]. Preclinical data by Amgen and UCB obtained on animal models did not identify an association either between an increased cardiovascular risk and sclerostin inhibition or a mechanism that could explain it [56]. Nevertheless, two *SOST* genetic variants were reported to be associated with a higher risk of myocardial infarction, coronary revascularization and major adverse cardiovascular events, as well as with some cardiovascular risk factors such as type 2 diabetes mellitus, high blood pressure, and central adiposity [57]. Some studies have found a correlation between high sclerostin levels and cardiovascular mortality [58], but the precise role of this elevation remains to be determined. Of note, the patients included in the ARCH study were older and had more prevalent vertebral fractures than those included in the phase 2 trial. They might therefore carry more comorbidities and cardiovascular risk factors. Observation of an increased cardiovascular risk in the ARCH study should also be tempered by the fact that it was not designed to compare the cardiovascular risk of the two treatments and a small number of events were observed [59], it is possible that the difference observed could be due to chance [60]. In men, however, an increased cardiovascular risk was also observed in the BRIDGE study, compared to the group receiving placebo. Men with serious cardiovascular events were more likely to have cardiovascular risk factors at baseline and were less likely to receive cardioprotective therapy [31]. Post-marketing survey of romosozumab in Japan, where romosozumab was launched in March 2019 without cardiovascular risk statement, raises concerns, as many more severe cardiovascular events including deaths were reported compared to teriparatide during the same time after commercialization [61]. It has to be kept in mind, though, that this kind of data are hard to compare, that spontaneous reports cannot prove causality and that more frequent reporting might have been encouraged by publicity around the potential cardiovascular risk of the drug.

A meta-analysis was recently conducted in order to better assess the cardiovascular risk with romosozumab. Six studies and 12,219 elderly men and postmenopausal women with osteoporosis were analyzed over a 12-month period [59]. It concluded that romosozumab does not increase the risk of cardiovascular events, including stroke, atrial fibrillation, heart failure and coronary artery disease (1.26 [95% CI 0.95–1.68], *p* = 0.11). The risk of 3P MACE (3-point major adverse cardiovascular event, grouping together death, myocardial infarction and stroke) was not significantly increased either (1.41 [95% CI 0.99–2.02], *p* = 0.06), but the risk of 4P MACE (3P MACE and heart failure) was increased (1.39 [95% CI 1.01–1.90], *p* = 0.04). However, after sensitivity analysis by random effects model, this was no longer significant (1.36 [95% CI 0.99–1.87], *p* = 0.06), meaning that the result should be interpreted cautiously. Romosozumab did not increase the risk of a single cardiovascular event, including that of aortic dissection, aortic valve disease or hypertension. There is an obvious need for trials designed to evaluate the cardiovascular risk of romosozumab and analyses of post-marketing data in order to better evaluate the risk.

## 7. Transition from Anti-Sclerostin Therapy at Treatment Cessation

Like denosumab and unlike bisphosphonates, romosozumab has no persistent effect, and benefits obtained under treatment are progressively reversible when stopped. It is therefore crucial to organize a transition to another anti-osteoporosis treatment after romosozumab discontinuation.

The FRAME study was designed with a transition of both groups to denosumab in the second year [9] and results of the extension study are now available after 2 years of denosumab, i.e., 3 years of treatment in total [62]. BMD continued increasing to reach at two years in the romosozumab-to-denosumab and placebo-to-denosumab groups respectively 17.6% and 5.0% at the lumbar spine, 8.8% and 2.9% at the total hip, 6.6% and 0.6% at the femoral neck. At 3 years, the differences in relative BMD gain between the groups remained similar, the benefit obtained with romosozumab during the first 12 months was maintained. After 2 years of transition to denosumab, relative risk reduction for the group initially on romosozumab was 66% for vertebral fractures, 27% for clinical fractures and 21% for non-vertebral fractures.

A case study was carried out in a subgroup of patients from the extension of the FRAME study, who were transitioned to oral (*n* = 5) or IV (*n* = 11) bisphosphonates at the end of trial, or who did not want treatment (*n* = 3). Zoledronate was administered within a median of 65 days and up to 6 months after the end of the study in order to increase its incorporation into the bone matrix and thus its effectiveness [63]. One year after the end of study, zoledronate allowed maintaining 73% [95% CI 61–85%] of BMD gain at the lumbar spine, and 87% [95% CI 77–98%] at the total hip. In contrast, patients who received no treatment lost between 80 and 90% of the benefit 12 months after the end of the study. The loss was intermediate for patients who received risedronate (35 mg/week). In the ARCH study [49], patients who received romosozumab or alendronate in the first year were all switched to alendronate for at least another 12 months. During the second treatment period, the benefit of romosozumab compared to alendronate was maintained. In both groups, there was a slight increase in BMD at the spine and hip. The risk of new vertebral fracture after 24 months was reduced by 48% in patients who had received romosozumab during the first year (6.2% vs. 11.9% for alendronate, *p* <0.001), the risk of clinical fracture was reduced by 27% (9.7% vs. 13%, *p* < 0.001), that of non-vertebral fracture by 19% (8.7% vs. 10.6%, *p* = 0.04) and that of hip fracture by 38% (2% vs. 3.2%, *p* = 0.02). The advantage acquired during the first year of treatment with a sclerostin inhibitor was therefore maintained with bisphosphonates.

Rather than a transition after 12 months, some investigated the effect of romosozumab for an additional year. A 24-month treatment with romosozumab was evaluated in a phase 2 study (*n* = 364), followed by one year of denosumab or placebo [26]. After 24 months of treatment, romosozumab promoted a gain in BMD of 15.1% at the lumbar spine and 5.4% at the total femur. Patients who received denosumab from months 24 to 36 had an average BMD gain of 19.4% in the lumbar spine and 7.1% in total femur at the end of the study, while those who received placebo had BMD values close to baseline. Romosozumab administered for 24 months was well tolerated. Thereafter, the effect of a second treatment sequence with romosozumab, after the 12-month period without treatment or with treatment with denosumab was evaluated in 35 patients [64]. In the 19 patients who had received placebo in the 3rd year, an additional year of romosozumab led to a new gain in BMD at the lumbar spine and total hip, similar to that obtained in the first year of treatment with romosozumab. In those who had received denosumab in the third year, the second treatment sequence with romosozumab resulted in an average gain in BMD at the lumbar spine of 2.3% and hip BMD was maintained. The safety profile during this second treatment sequence was similar to that observed during the first one. A single dose of zoledronate administered approximately 4 weeks after the last dose of romosozumab allowed preservation of BMD gains for up to 2 years after the second sequence [65].

## 8. Anti-Sclerostin Therapy in Other Bone Diseases

Given the effectiveness of anti-sclerostin antibodies in reducing fracture risk in osteoporosis, trials have been initiated in other bone diseases.

Anti-sclerostin antibodies were able to improve bone strength and microarchitecture, and to reduce axial and long bone fractures in animal models of osteogenesis imperfecta (OI) [66,67]. An anti-sclerostin antibody (Setrusumab, BPS 804) has thereafter been tested in a clinical phase 2a trial in 14 adult patients with moderate OI. Three increasing doses administered every 2 weeks allowed a marked increase in markers of bone formation, along with a decrease in CTX levels and a BMD gain at the lumbar spine (+ 4%). The treatment was well tolerated [10]. A phase 2b trial including 112 adults with Type I, III, and IV OI is in progress. Administration of setrusumab for 12 months resulted in an increase in BMD at the lumbar spine of 8.8% in the highest dose group compared to baseline [68]. Improvement was observed in all OI subtypes, along with a trend towards the incidence of fractures. The tolerance was good, without any occurrence of cardiovascular event. The study is ongoing with a 12-month extension phase during which patients receive zoledronic acid or do not receive treatment. A phase 3 trial is planned in a pediatric population of osteogenesis imperfecta.

A phase 2a study in 8 patients was also conducted in hypophosphatasia with positive results in terms of bone markers and BMD gain at the lumbar spine, but development has not yet been continued in this indication [69]. The rationale of the anti-sclerostin therapy in this condition remains to be justified, since it is not likely to improve the defective matrix mineralization characterizing hypophosphatasia. It was suggested that anti-sclerostin antibodies could also be useful in the treatment of multiple myeloma. Pre-clinical studies have shown that anti-sclerostin antibodies could increase the number of osteoblasts, the rate of bone formation, and thus reduce lytic bone damage. Combination of anti-sclerostin antibodies and zoledronic acid allowed an increase in bone mass and a reduction of the fracture rate. However, anti-sclerostin antibodies showed no effect on tumor burden [70]. A preclinical study suggested that anti-sclerostin antibodies may be beneficial in the treatment of musculoskeletal complications of breast cancer [71]. In a murine model, anti-sclerostin antibodies reduced bone metastasis, prevented bone destruction and improved muscle function and survival. More controversial data have emerged in rheumatoid arthritis, some preclinical studies suggesting that the inhibition of sclerostin could prevent the development of bone erosions or bone loss, while other suggested that sclerostin could play a protective role [21]. Sclerostin could also be protective, and therefore its inhibition deleterious, in osteoarthritis as well as in ankylosing spondylitis [21]. In the latter condition, serum sclerostin levels are lowered and patients with the lowest levels are more likely to develop syndesmophytes and ankylosis [72]. However, these results remain to be confirmed in larger human studies.

## 9. Conclusions

Anti-sclerostin antibodies are interesting therapies in the prevention of fractures as they enable concomitant stimulation of bone formation and inhibition of bone resorption, at least during the first months of treatment. In particular, romosozumab has shown its effectiveness in the treatment of postmenopausal and male osteoporosis, with good tolerance in short-term trials. Concerns remain about a possible increase in serious cardiovascular events. This treatment also presents interesting performance against other anti-osteoporosis treatments available, thanks to its rapid effectiveness related to its mode of action. It has thus been recently approved in several countries including Europe and the US, for the treatment of severe osteoporosis in postmenopausal women at high risk of fractures, for a period of 12 months. Currently, romosozumab is generally contraindicated in case of a history of myocardial infarction or stroke. Given that the effect is reversible when discontinued, romosozumab requires a transition to an anti-resorbing agent at the end of treatment. Further studies should in particular allow a better assessment of the risk and the place of this treatment in the therapeutic panel against osteoporosis, as well as the most suitable therapeutic sequences according to patients’ profile. Anti-sclerostin antibodies may also find a place in the future in the treatment of other bone diseases.

## Figures and Tables

**Figure 1 jcm-09-03439-f001:**
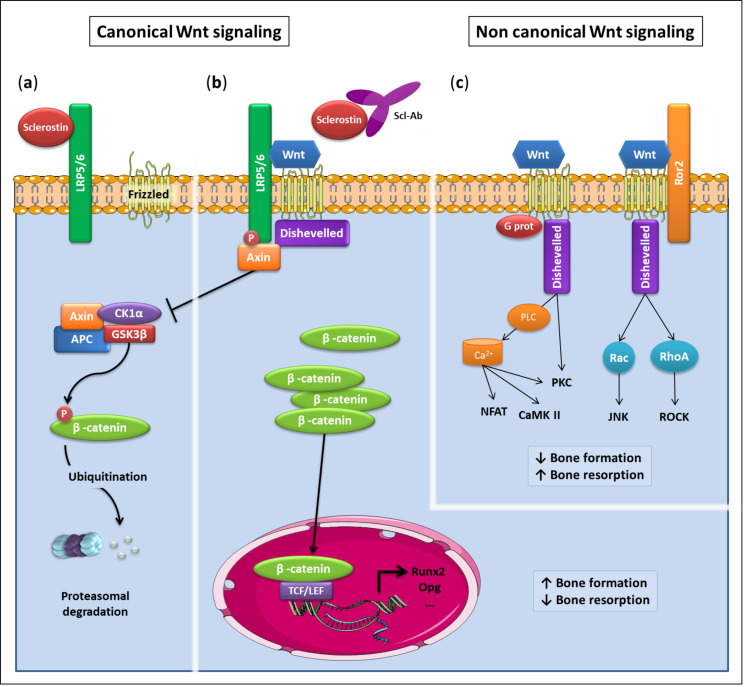
Canonical and non-canonical Wnt signaling pathways. (**a**) Wnt signaling OFF: If there is no Wnt ligand, or if sclerostin prevents its binding to the receptor complex, the destruction complex Axin-APC-CK1α-GSK3 phosphorylates β-catenin, targeting it for ubiquitination and degradation by the proteasome. (**b**) Wnt signaling ON: Canonical Wnt signaling is activated by Wnt ligands binding to Frizzled receptors and LRP5/6 co-receptors, resulting in the recruitment of axin and disheveled by the receptor complex and leading to GSK3 inhibition, the mechanism of which is still debated. β-catenin accumulates, translocates into the nucleus and associates with transcription factors to induces the expression of target genes. Scl-Ab blocks the action of sclerostin, preventing its binding to Lrp5/6 and therefore canonical Wnt signaling inhibition. (**c**) Non-canonical Wnt signaling can also be activated by binding of Wnt ligands to Frizzled receptors. In the Wnt/Ca2+ pathway, increased intracellular calcium concentration, possibly via G proteins and PLC, activates CaMKII, PKC and NFAT. In the Wnt/planar cell polarity (PCP) pathway, small G proteins such as Rac and Rho are activated, resulting in JUNK and ROCK activity. APC: APC adenomatous polyposis coli; CaMKII: calmodulin-dependent protein kinase type 2; CK1α: Casein kinase 1 α; GSK3: Glycogen synthase kinase 3; JUNK: Jun kinase; LRP: Low-density lipoprotein receptor-related protein; NFAT: Nuclear factor of activated T cells; PKC: Protein kinase C; PLC: phospholipase C; ROCK: RHO-associated kinase; ROR: Receptor-tyrosine-kinase-like orphan receptor; Scl-Ab: Anti-sclerostin antibody.

**Table 1 jcm-09-03439-t001:** Action of different anti-osteoporosis treatments on bone.

Treatment	Therapeutical Class	Type of Treatment	Effect on Bone Modeling and Remodeling	Overall Bone Formation/Resorption	Ob Number /Activity	Oc Number /Activity
**Alendronate**	Bisphosphonate	Anti-resorptive	↓ remodeling ? modeling	↓/↓	↔/↔	↓/↓
**Denosumab**	Anti-RANKL antibody	Anti-resorptive	↓ remodeling ↔modeling [40]	↓/↓	↔/↔	↓/↓
**Teriparatide**	Recombinant human PTH	Anabolic	↑ remodeling ↑ modeling (transient) [41]	↑/↑	↑/↑	↑/↑
**Romosozumab**	Anti-sclerostin antibody	Anabolic	↑ modeling (transient) ↓ remodeling	↑ then ↓/↓	↑ then ↓/↑	↓/↓

Ob: osteoblast; Oc: osteoclast; ↑: increased; ↓: decreased; ↔: stable; ?: no data available.

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
