# Peer review of "Anti-Sclerostin Antibodies in Osteoporosis and Other Bone Diseases"

_jcm, 2020, doi:10.3390/jcm9113439_

Round 1

Reviewer 1 Report

Fabre et al. discussed on the role of sclerostin and their efficacy/safety as well as a comparison with the other drugs in bone diseases including osteoporosis. Identification of therapeutic targets in osteoporosis is an important topic so we need a better understanding of the cellular/molecular mechanism(s) involved in the entire Wnt signaling pathway in skeletal cells and their role in bone disorders. The manuscript is clearly written, informative, and the approaches to make their work-flows seem appropriate.

To this reviewer, it would be great for the readers if the authors could describe each drugs with their phase, target cells or mechanisms (eg. Decreased bone resorption, increased bone formation, or both; osteoclast number/activation or osteoblast number/activation), target molecules, and each references in a table. Although the authors well-established the Wnt signaling pathways and the known target genes in the figure, there is lack of explanations in the manuscript and this reviewer is still wondering if this figure is necessary for the publication. Again, they should update more explanation on the figure and add any tables to show the actions of those drugs with convincing information in the manuscript.

Minor comments.

  1. The purpose of this review manuscript is missing in the Abstract.
  2. Is there any limitation of the usage of DKK1 antagonists? If so, please describe it because this molecules are also involved in Wnt signaling pathway.
  3. There are too many paragraphs in section 6.1 and 8.

Author Response

  • It would be great for the readers if the authors could describe each drugs with their phase, target cells or mechanisms. Thank you for this very interesting suggestion. Here is the table we added to the review (Table 1), lines 140 and 154

Treatment

Therapeutical class

Type of treatment

Effect on bone modeling and remodeling

Overall bone formation / resorption

Ob number / Activity

Oc number / Activity

Alendronate

Bisphosphonate

Anti-resorptive

↓remodeling

? modeling

↓ / ↓

↔ / ↔

↓ / ↓

Denosumab

Anti-RANKL antibody

Anti-resorptive

↓remodeling

↔ modeling (1)

↓ / ↓

↔ / ↔

↓ / ↓

Teriparatide

Recombinant human PTH

Anabolic

↑remodeling

↑modeling (transient) (2)

↑ / ↑

↑ / ↑

↑ / ↑

Romosozumab

Anti-sclerostin antibody

Anabolic

↑modeling (transient)

↓remodeling

↑ puis ↓ / ↓

↑ puis  ↓ / ↑

↓ / ↓

  1. Dempster, D.W.; Zhou, H.; Recker, R.R.; Brown, J.P.; Recknor, C.P.; Lewiecki, E.M.; Miller, P.D.; Rao, S.D.; Kendler, D.L.; Lindsay, R.; et al. Remodeling- and Modeling-Based Bone Formation With Teriparatide Versus Denosumab: A Longitudinal Analysis From Baseline to 3 Months in the AVA Study. J Bone Miner Res 2018, 33, 298–306, doi:10.1002/jbmr.3309.
  2. Dempster, D.W.; Zhou, H.; Ruff, V.A.; Melby, T.E.; Alam, J.; Taylor, K.A. Longitudinal Effects of Teriparatide or Zoledronic Acid on Bone Modeling- and Remodeling-Based Formation in the SHOTZ Study. J Bone Miner Res 2018, 33, 627–633, doi:10.1002/jbmr.3350.
  • Although the authors well-established the Wnt signaling pathways and the known target genes in the figure, there is lack of explanations in the manuscript. We also added some explanations about the Wnt pathway signaling in the main text lines 56-63: “In the absence of Wnt ligand, a destruction complex phosphorylates beta-catenin, targeting it for ubiquitination and degradation by the proteasome. When Wnt ligands bind to Frizzled receptor and LRP5/6 (low density lipoprotein receptor-related protein 5 or 6) co-receptor, there is an inhibition of the destruction complex. Beta-catenin accumulates, translocates into the cell nucleus and triggers the transcription of genes involved in bone formation (Figure 1b). Non-canonical Wnt pathways stimulating bone formation can also be activated by Wnt ligands but they increase bone resorption [3] (Figure 1c).”
  • The purpose of this review manuscript is missing in the Abstract. The purpose of the review has been added in the introduction lines 22-24: “This review aims to outline the role of sclerostin, the efficacy and safety of anti-sclerostin therapies and in particular romosozumab and their place in therapeutic strategies against osteoporosis or other bone diseases.”
  • Is there any limitation of the usage of DKK1 antagonists? If so, please describe it because this molecules are also involved in Wnt signaling pathway. Anti-DKK1 antibodies are not currently developed in osteoporosis. A phase 1b study in multiple myeloma did not report any serious adverse event related to treatment. Phase 2 study results have not been published, but the study did not evidence any serious adverse event related to treatment, adverse events are not detailed.
  • There are too many paragraphs in section 6.1 and 8. Paragraphs have been regrouped in sections 6.1 and 8

Reviewer 2 Report

Dear Authors,

please address the following issues.

  • there are multiple missing references in the introduction. There are parts considered a general knowledge, ans these indeed can be spared the references, otherwise appropriate citation is necessary
  •  when discussing anti-fracture effects there are studies comparing romosozumab with alendronate new report on ARCH study could be mentioned: Lau EMC, Dinavahi R, Woo YC, Wu CH, Guan J, Maddox J, Tolman C, Yang W, Shin CS. Romosozumab or alendronate for fracture prevention in East Asian patients: a subanalysis of the phase III, randomized ARCH study. Osteoporos Int. 2020 Apr;31(4):677-685. doi: 10.1007/s00198-020-05324-0. Epub 2020 Feb 11. PMID: 32047951; PMCID: PMC7075830.

Author Response

  • There are multiple missing references in the introduction.The authors thank you for your comments on references. Additional references have been added in the introduction for more accuracy (references 1 to 11, lines 34 to 42).
  • When discussing anti-fracture effects there are studies comparing romosozumab with alendronate new report on ARCH study could be mentioned. The subanalysis of the ARCH study in the East-Asian population has been mentioned in the paragraph 5.2 p6 line 216-217: “A subanalysis conducted among the ARCH East-Asian population on 275 patients showed similar outcomes [48]”

Reviewer 3 Report

This review summarize the actual findings about the anti sclerostin therapy for osteoporosis and other bone metabolic bone diseases. Nevertheless recently other authors published similar reviews, this paper is methodologically accurate and complete and also the reference list is up to date. It covers all the results of clinical trials about the use of romosozumab in osteoporotic patients and also focus the issue about treatment transition and adverse events. Moreover the authors indicate the results of some papers about the use of this treatment in other metabolic bone diseases.

Just a minor comment:

the title of paragraph 3 could be changed in “Efficacy of romosozumab in osteoporotic women and men” and could be moved after the section about mechanism of action for a better narrative sequence.

Author Response

The title of paragraph 3 could be changed in “Efficacy of romosozumab in osteoporotic women and men” and could be moved after the section about mechanism of action for a better narrative sequence.

Thank you for your comments, we changed the title of paragraph 3 as suggested, line 76. Even if a different order of sections could also be nice and relevant, we would prefer not to change it as we think it is interesting to start from clinical data and then to explore how these result can be obtained.